# Cardiac-specific troponins in uncomplicated pregnancy and pre-eclampsia: A systematic review

Samuel Dockree[1]*, Jennifer Brook[2], Brian Shine[2], Tim James[2], Lauren Green[1], Manu Vatish[3]

1 Women's Centre, John Radcliffe Hospital, Oxford University Hospitals NHS Foundation Trust, Oxford, United Kingdom, 2 Department of Clinical Biochemistry, John Radcliffe Hospital, Oxford University Hospitals NHS Foundation Trust, Oxford, United Kingdom, 3 Nuffield Department of Women's and Reproductive Health, University of Oxford, Oxford, United Kingdom

* samuel.dockree@ouh.nhs.uk

## Abstract

### Background

The risk of myocardial infarction (MI) increases during pregnancy, particularly in women with pre-eclampsia. MI is diagnosed by measuring high blood levels of cardiac-specific troponin (cTn), although this may be elevated in women with pre-eclampsia without MI, which increases diagnostic uncertainty. It is unclear how much cTn is elevated in uncomplicated and complicated pregnancy, which may affect whether the existing reference intervals can be used in pregnant women. Previous reviews have not investigated high-sensitivity troponin in pregnancy, compared to older, less sensitive methods.

### Methods

Electronic searches using the terms "troponin I" or "troponin T", and "pregnancy", "pregnancy complications" or "obstetrics". cTn levels were extracted from studies of women with uncomplicated pregnancies or pre-eclampsia.

### Results

The search identified ten studies with 1581 women. Eight studies used contemporary methods that may be too insensitive to use reliably in this clinical setting. Two studies used high-sensitivity assays, with one reporting an elevation in troponin I (TnI) in pre-eclampsia compared to uncomplicated pregnancy, and the other only examining women with pre-eclampsia.

Seven studies compared cTn between women with pre-eclampsia or uncomplicated pregnancy using any assay. Seven studies showed elevated TnI in pre-eclampsia compared to uncomplicated pregnancy or non-pregnant women. One study measured troponin T (TnT) in pregnancy but did not examine pre-eclampsia.

**Data Availability Statement:** All relevant data are within the manuscript and its Supporting information files.

**Funding:** The authors received no specific funding for this work.

**Competing interests:** The authors have declared that no competing interests exist.

## Conclusion

TnI appears to be elevated in pre-eclampsia, irrespective of methodology, which may reflect the role of cardiac stress in this condition. TnI may be similar in healthy pregnant and non-pregnant women, but we found no literature reporting pregnancy-specific reference intervals using high-sensitivity tests. This limits broader application of cTn in pregnancy. There is a need to define reference intervals for cTn in pregnant women, which should involve serial sampling throughout pregnancy, with careful consideration for gestational age and body mass index, which cause dynamic changes in normal maternal physiology.

## Introduction

Cardiac-specific troponins (cTn) are important circulating biomarkers used for investigating suspected myocardial ischaemia in modern clinical practice, and their elevation is an essential criterion for the diagnosis of acute myocardial infarction (MI) [1]. Myocardial damage is multifactorial, and the release of cTn from injured cells may result from myocardiocyte apoptosis, with or without overt evidence of necrosis [2]. Interpretation of cTn results is achieved through comparison to the 99th percentile of a well-characterised reference population. Two troponin markers are used in clinical practice: troponin I (TnI) and troponin T (TnT). Whilst these are distinct proteins, both are used as they both reflect myocyte injury and have similar diagnostic accuracy for MI [3]. A review of 417 healthy non-pregnant women aged 19–91 reported non-parametric 99% upper reference limits of 3–40 ng/L for TnI, and 10–12 ng/L for TnT, depending on the assay used [4]. While these limits were not affected materially by age, they were substantially higher in men than women by a median factor of 1.54 (range 0.9–3.2) for TnI, and 1.55 (1.17–1.6) for TnT, and the advantages of using sex-specific reference intervals for cTn for predicting major adverse cardiac events are well-documented [5]. Importantly, the sample bank from which these intervals were derived did not enquire about pregnancy [6].

Cardiovascular disease remains the leading cause of maternal mortality in the UK, constituting almost one in four deaths in women during or immediately following pregnancy [7]. Pregnant women are three to four times more likely to suffer an MI than non-pregnant women, with a much higher incidence in those older than 40 years [8]. A history of coronary artery disease is associated with hypertensive disorders of pregnancy, such as pre-eclampsia [9], which are in turn associated with an increased risk of MI (OR 1.6, 95% CI 1.0–2.5) [10]. Physiological changes of pregnancy include an increased cardiac output with left ventricular remodeling [11] and diastolic dysfunction [12], prompting speculation that cTn may be elevated in women with uncomplicated pregnancy. It has also been proposed that cTn may be further elevated in women with pre-eclampsia [13], a multi-system hypertensive disorder of pregnancy that is also independently associated with altered myocardial hypertrophy and exacerbated diastolic dysfunction [14].

The earliest routinely used methods for measuring cTn, sometimes referred to as first generation, could only detect the abnormally high levels of TnI such as those seen in extensive myocardial necrosis (above 10,000 ng/L), using radioimmunoassays on post-mortem cardiac tissue [15]. Progressive technological developments have improved the sensitivity by improving precision and allowing for earlier detection, but even assays described as fourth generation could only detect circulating cTn in a relatively small proportion of the healthy population, with quoted detection limits between 2–200 ng/L [16,17]. Therefore, these older methods are

unable to detect the nuanced effects of micro-necrosis. The recent development of fifth generation cTn tests, which are 100–1000 times more sensitive than their predecessors [18], has allowed accurate measurement of much lower concentrations than was previously possible, down to 0.1–5 ng/L [16,17]. By definition, high-sensitivity cTn tests can measure cTn above the lowest detectable limit in at least half of the healthy population, with a coefficient of variation below 10% at the 99th percentile [16]. By measuring very low levels of cTn, one may extend its utility from solely identifying individuals with MI to understanding temporal changes within individuals, and investigating associations with abnormal physiological states or non-cardiac disease, such as pregnancy or pre-eclampsia. There are no established reference intervals for cTn in pregnant women, despite the significant physiological changes observed in pregnancy, and previous reviews on cTn in pregnancy have not considered the use of new, highly sensitive methods independently [13]. To address this, we performed a systematic review to investigate cTn in studies of women with uncomplicated pregnancy or pre-eclampsia, using both new and older testing methods.

## Methods

### Protocol and registration

This review was prospectively registered with PROSPERO (CRD42020178159) and was undertaken according to PRISMA guidelines (Preferred Reporting Items for Systematic Reviews and Meta-Analyses) [19] (see S1 Table).

### Eligibility criteria

Published studies of any size, investigating levels of cTn in women with either uncomplicated pregnancy or pre-eclampsia, using the following PECOS definitions:

Population (P): adult pregnant women with a live pregnancy of any gestation before the onset of labour, including women of any ethnicity or nationality;

Exposure (E): pre-eclampsia, using any definition;

Comparison (C): uncomplicated pregnancy;

Outcome (O): levels of cTn (ng/L) and reference intervals used;

Study design (S): observational studies (prospective or retrospective) or randomised control trials.

We excluded studies solely of women in labour or after delivery, miscarriage or termination of pregnancy, studies where levels or reference intervals for cTn were not reported, conference abstracts and reviews.

### Sources, search strategy & screening

Electronic searches of Medline (1946-September 2020) and Embase (1974-September 2020), including MeSH terms, including all subheadings and as a keyword search. Results were restricted to studies in English, with human participants. The search strategy, including all keywords, is presented in S2 Table.

We screened references from potentially eligible studies and reviews for other papers by hand. If studies were not available electronically, we contacted authors once by email and excluded them if there was no response. Two obstetricians (SD, MV) independently checked the abstracts and full texts of potential studies against the inclusion criteria.

## Data extraction

We extracted data on levels of cTn (ng/L) reported in uncomplicated pregnancy and/or pre-eclampsia, the reference intervals used, the study design and primary outcomes, and demographic data of the populations studied. Discrepancies were mediated by discussion (LG). We recorded data on clinically relevant background characteristics and presented the mean and standard deviation unless otherwise stated. If the mean was not reported the median was extracted with an acknowledgment that it may differ from the mean depending on the underlying distribution.

## Risk of bias & certainty of evidence

We assessed the quality of included studies and the risk of bias using preset criteria (see S3 Table) where observational studies began as "low" quality and were upgraded or downgraded according to their strengths and weaknesses [20]. An assessment of the certainty of the evidence was summarised using the GRADE tool [21], where the outcomes in question were "the association between levels of cTn and uncomplicated pregnancy" and "the association between levels of cTn and pre-eclampsia" (see S4 Table).

# Results

## Study selection and characteristics

The initial search identified 616 citations, excluding duplicates. The search strategy is presented in fully in S2 Table. From this list we identified ten published, peer-reviewed papers reporting levels of cTn in 1581 pregnant women with uncomplicated pregnancy or pre-eclampsia. The frequencies and reasons for exclusion are presented in Fig 1.

Of the ten studies included in this review, nine studies investigated TnI and one study investigated TnT. One study was excluded as it did not consider women with pre-eclampsia separately to those with an uncomplicated pregnancy [22]. Two studies reported cTn only in uncomplicated pregnancy (one TnI [23] and one TnT [24]) and one study reported TnI only in women with pre-eclampsia [25]. The median number of women in each study was 73.5 (range 40–880), with a median maternal age of 29 (range 24–31).

## Results of individual studies

The demographic data and main findings for studies of earlier generation cTn tests and highly sensitive TnI are presented in Tables 1 and 2, respectively. The majority of studies used earlier generation tests and lacked sufficient method descriptions or performance characteristics to allow comparison of limits of detection and quantification, which are both critical to determine whether methods have adequate sensitivity to detect changes to cTn that may occur in pregnancy. Two studies [25,26] defined the method with sufficient detail to enable categorization against the IFCC (International Federation of Clinical Chemistry) classification as highly sensitive [27]. One study compared healthy pregnant and non-pregnant women, reporting no difference in TnI [28]. This study was included in the review as it also recruited healthy pregnancy women, but the search was not designed to identify studies solely of non-pregnant individuals. In the absence of a pregnancy-specific reference interval, the primary focus of the remaining studies in this review was the comparison between women with uncomplicated and complicated pregnancies (those with pre-eclampsia).

**cTn using contemporary (non-highly sensitive) methods.** In the studies that compared TnI between groups, the median TnI was 30 ng/L (range 10–300 ng/L) in uncomplicated pregnancy and 460 ng/L (range 155–1020 ng/L) in pre-eclampsia. Fleming, *et al*. reported a

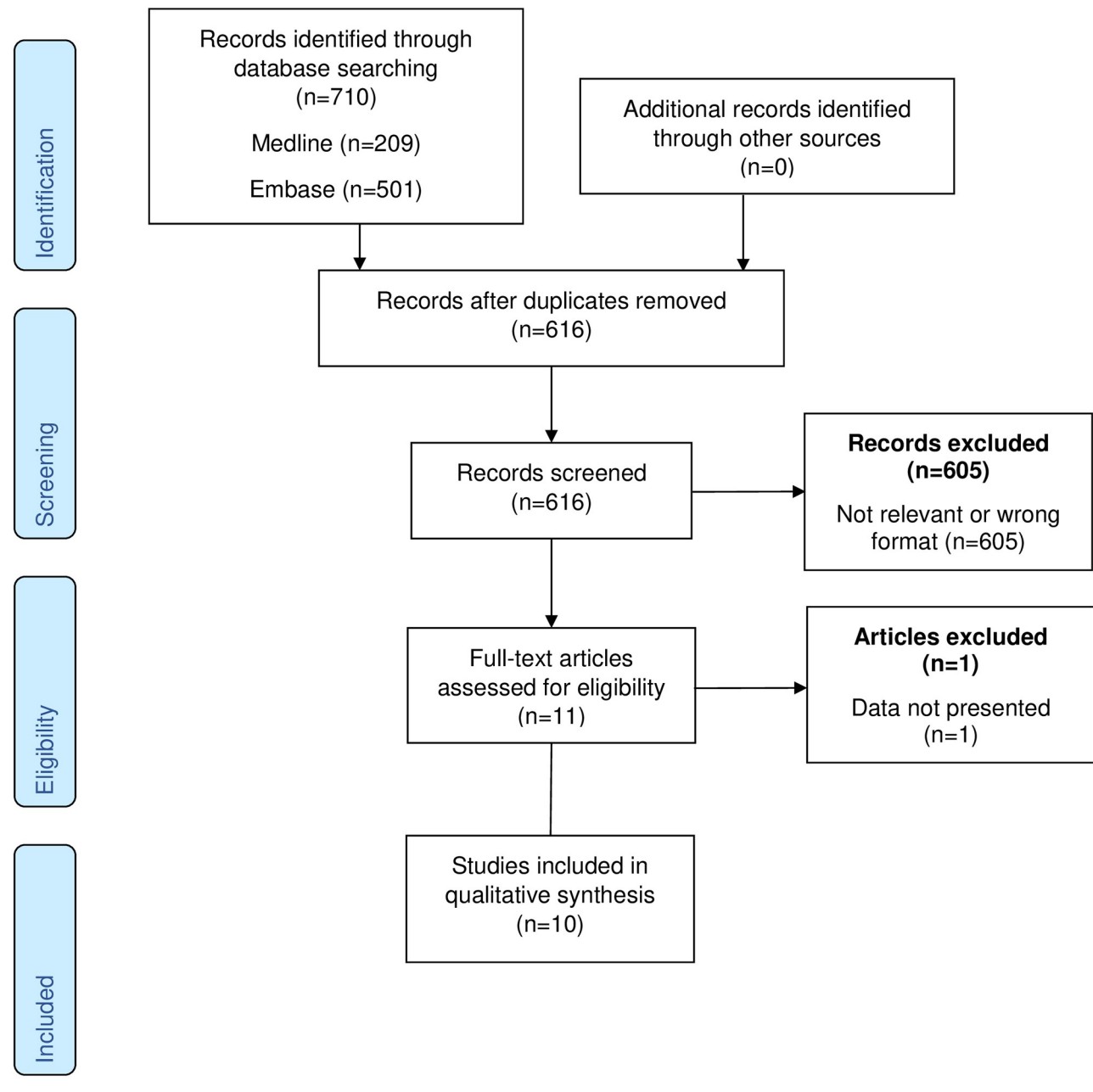

**Fig 1. PRISMA flowchart.**

statistically significant increase in TnI in women with gestational hypertension compared to normotensive women (median 89 *vs*. 30 ng/L, p<0.001) and a subsequent increase in women with pre-eclampsia compared to gestational hypertension (155 *vs*. 89 ng/L, p = 0.03). The one study investigating TnT in uncomplicated pregnancy reported all normal values below the upper cut-off quoted by its manufacturer [24]. However, the demographics of the population from which this reference interval is derived is unclear, as is the percentile determining its upper limit [34]. It is uncertain whether this population included men, which may elevate the

**Table 1. Studies of cardiac troponins using contemporary assays[a].**

| Study | Population (I: inclusion) (E: exclusion) | Size (n) (UP) (PET) | Manufacturer—instrument | Age[b] (years) | Demographics | Main findings | cTn reference interval (ng/L) | cTn in UP (Mean (SD); ng/L) | cTn in PET (Mean (SD); ng/L) | P_diff |
|---|---|---|---|---|---|---|---|---|---|---|
| | | | | **Troponin T** | | | | | | |
| Adamcova, et al. (1999) [24] | **I**: women UP or undergoing tocolysis at 32–36 weeks. | 42 (20) (-) | Boehringer Mannheim—Enzymun | 24 | Age, weight, height, parity, multiple pregnancy, gestation. | TnT elevated with tocolysis. | ≤100 | 80 (10) | - | - |
| | | | | **Troponin I** | | | | | | |
| Fleming, et al. (2000) [29] | **I**: women with UP, GH or PET. | 69 (43) (6) | Beckman—Access | 29 | Age, gestation, parity, MAP. | TnI elevated in PET. | ≤30 | 30[c] | 155[c] | 0.002 |
| Atalay, et al. (2005) [30] | **I**: women with UP or PET. **E**: other hypertension, diabetes, cardiac disease, other obstetric complications. | 58 (28) (30) | Beckman—Access | 26 | Age, parity, gestation, BP, proteinuria. | TnI elevated in PET. | - | 20 (0–50)[c] | 200 (20–4530)[c] | <0.001 |
| Aydin, et al. (2009) [31] | **I**: women with severe PET or no hypertensive disorder. **E**: mild PET, infection, diabetes, premature membrane rupture, oligo/polyhydramnios, heart/renal disease, hypertension, inotropic therapy. | 78 (42) (36) | Siemens—Immulite | 24 | Age, parity, gestation, fetal weight, Apgar score, BP, proteinuria. | TnI not elevated in severe PET. | <40 | 300 (100) | 660 (1730) | 0.886 |
| Rafik Hamad, et al. (2009) [32] | **I**: nulliparous women with UP or PET. **E**: smokers, antihypertensive therapy, chronic disease, IVF or egg donation, multiple pregnancy, extreme obesity. | 65 (30) (35) | Abbott—Architect Ci8200 | 31 | Age, weight, height, history of hypertension, BP, gestation, body surface area, gestation and birthweight at delivery. | TnI not elevated in PET. | <22 | All <22 | All <22 | - |
| Pasupathi, et al. (2010) [28] | **I**: women with UP or PET, age 19–37 and non-pregnant women. | 150 (50) (50) | Abbott—Axsym | 27 | Age, weight, BMI, hypertension, diabetes, BP, gestation. | TnI not elevated in UP but elevated in PET. | - | 10 | 1020 (93) | <0.001 |
| Ersoy, et al. (2016) [23] | **I**: women with UP or placenta praevia. **E**: multiple pregnancy, PPROM, previous complicated pregnancy, uterine surgery, thyroid/renal disease, hypertension, epilepsy, diabetes, use of alcohol/cocaine/cardiac drugs. | 100 (46) (-) | Beckman—Access 2 | 30 | Age, BMI, gravidity, parity, history of dilatation and curettage. | TnI elevated in placenta praevia. | - | 3 (0.9) | - | - |
| Ekun, et al. (2018) [33] | **I**: women with UP or PET, >20 weeks gestation. **E**: malaria, anaemia, hepatitis, diabetes, renal failure, tuberculosis, autoimmune disease. | 99 (50) (49) | Not reported (described as "ELISA") | - | - | TnI elevated in PET. | - | 130 (140) | 460 (310) | <0.001 |

[a]Methods are classified as either contemporary for the period of study or as high sensitivity against the IFCC definition [27].

[b]For all women or the largest group reported (mean or median).

[c]Median ± (minimum-maximum).

Abbreviations: Cardiac-specific troponin (cTn); Troponin I (TnI); uncomplicated pregnancy (UP); gestational hypertension (GH); pre-eclamptic toxaemia (PET); in-vitro fertilisation (IVF); body mass index (BMI); protein:creatinine ratio (PCR); mean arterial pressure (MAP), preterm premature rupture of membranes (PPROM), blood pressure (BP).

**Table 2. Studies of cardiac troponins using high-sensitivity assays[a].**

| Study | Population (I: inclusion) (E: exclusion) | Size (n) (UP) (PET) | Testing method | Age[b] (years) | Demographics | Main findings | cTn reference interval (ng/L) | cTn in UP (Mean (SD); ng/L) | cTn in PET (Mean (SD); ng/L) | P$_{diff}$ |
|---|---|---|---|---|---|---|---|---|---|---|
| | | | | **Troponin I** | | | | | | |
| Morton, *et al.* (2018) [25] | **I**: women with PET, 23–41 weeks. **E**: abnormal renal function, cardiac disease, chronic hypertension, obstructive sleep apnoea, illicit drug use, pulmonary embolism. | 40 (-) (40) | Abbott—Architect Stat | 29 | Age, gestation at delivery, gravidity, BMI, smoking, ethnicity, PCR, MAP. | Linear association between MAP and log TnI. | ≤15.6 | - | 20 | - |
| Ravichandran, *et al.* (2019) [26] | **I**: women with UP, GH or PET, age 18–35, gravida ≤5. **E**: hypertension, diabetes, haematological disorder, blood transfusion, personal/family history of cardiac disease. | 880 (842) (10) | Abbott—Architect Stat | 29 | Age, ethnicity, BMI, trimester, | TnI elevated in PET. | ≤16 | 1 (0–1)[cd] | 12 (3.0–97.5) [cd] | <0.001[e] |

[a]Methods are classified as either contemporary for the period of study or as high sensitivity against the IFCC definition [27].

[b]For all women or the largest group reported (mean or median).

[c]Median (interquartile range).

[d]Including postnatal women.

[e]Comparing any hypertension to normotensive women.

Abbreviations: Cardiac-specific troponin (cTn); Troponin I (TnI); uncomplicated pregnancy (UP); gestational hypertension (GH); pre-eclamptic toxaemia (PET); in-vitro fertilisation (IVF); body mass index (BMI); protein:creatinine ratio (PCR); mean arterial pressure (MAP), preterm premature rupture of membranes (PPROM), blood pressure (BP).

upper reference limit, although the reported limit (100 ng/L) is markedly higher than those reported by Apple, *et al.* for either men or women [4].

**TnI using highly sensitive methods.** Ravichandran, *et al.* showed very low levels of TnI in 842 uncomplicated pregnancies (median 1.0 ng/L, IQR 0.0–1.0 ng/L) and higher levels in women with pre-eclampsia (median 12.0 ng/L, IQR 3.0–97.5), when considering pregnancies of any gestation [35]. Morton, *et al.* used the same testing method and reported similarly elevated levels of TnI in women with pre-eclampsia (mean 20 ng/L).

## Data synthesis

The significant heterogeneity between testing methods, case validation and reporting procedures prevented us from making meaningful quantitative comparisons of cTn in pregnant women with and without pre-eclampsia. Instead, we made descriptive comparisons and summarised the findings further in the narrative GRADE assessment.

## Quality assessment & certainty of evidence

We graded one study as being of "moderate" quality [30] and all other studies were "low" or "very low". The quality assessment tool and findings are presented in S3 Table. The certainty of evidence for both outcomes was low (see S4 Table for the narrative). The studies included in the assessment of "The association between levels of cTn and uncomplicated pregnancy" mostly found that cTn was not elevated in uncomplicated pregnancy, although four of these

studies did not report a reference interval from which to draw comparisons. Six of the eight studies investigating "The association between levels of cTn and pre-eclampsia" reported elevated levels of cTn in women with pre-eclampsia, although only two of these were compared to a reference interval.

## Discussion

Overall, TnI was higher in women with pre-eclampsia than those with an uncomplicated pregnancy. Our findings are consistent with those of Pergialiotis, *et al.* [13] but our review considers both cardiac troponins and also includes some large, recent studies with clinically important findings. Notably, these studies used newer higher sensitivity assays [25,26], which is an important new investigation given that high-sensitivity cTn tests are now essential for diagnosing MI in modern clinical practice, and this has previously not been investigated in pregnancy. TnI in women with an uncomplicated pregnancy was similar to non-pregnant women but has rarely been compared directly. Pasupathi, *et al.* reported a mean TnI of 10 ng/L in women with uncomplicated pregnancy, which was not significantly different from healthy non-pregnant women (mean $<10$ ng/L, $P_{diff}>0.05$). This is consistent with the non-pregnant upper reference limits reported by Apple, *et al.* [4], although these results are on the lower detection limit for the assay used, so further inference cannot be made about the magnitude of this difference.

It would be clinically important to use modern technological developments in biochemistry to begin to understand if and how cTn changes during pregnancy, in the context of a rapidly expanding plasma volume and altered renal clearance, which are known to affect cTn [36]. With the advent of highly sensitive cTn tests it would now be possible to investigate gestation-specific changes within healthy individuals; Ravichandran *et al.* reported no association with gestational age but they did not differentiate between women with uncomplicated pregnancies and pre-eclampsia. These gestational age changes were not examined in the same women. Our working group has previously defined the pregnancy-specific reference interval for another biomarker, procalcitonin, in which we describe the importance of repeated measurements in each trimester, and we highlight the importance of investigating changes in biomarkers driven by plasma volume and BMI [37]. We propose that a study of cTn using similar methods, with longitudinal sample in each trimester, would allow for a more accurate comparison of cTn levels in uncomplicated and complicated pregnancy (e.g., pre-eclampsia), as well as with non-pregnant women.

TnI is independently associated with systolic blood pressure and left ventricular hypertrophy in non-pregnant adults [38]. Pre-eclampsia is also associated with cardiac hypertrophy and elevated systolic (and diastolic) pressure as well as diastolic dysfunction, suggesting that hypertension and cardiac strain may drive the elevation in TnI seen in pre-eclampsia, yet the data are not robust enough to conclude this. TnT was reported in only one small study using an earlier generation test, preventing us from drawing conclusions about TnT and pre-eclampsia from the existing literature.

One of the more significant recent developments improving the diagnosis of pre-eclampsia has been understanding the role of soluble fms-like tyrosine kinase 1 (sFlt) and placental growth factor (PlGF). These placentally-derived biomarkers have revolutionised the diagnosis of pre-eclampsia and may improve case validation in studies investigating hypertensive disorders of pregnancy [39]. While most of the papers in this review defined their case definitions, the diagnostic criteria varied and none of the papers used biomarkers, like sFlt or PlGF, to objectively refute or define pre-eclampsia and to distinguish it from other causes of hypertension.

## Strengths and limitations

We performed an extensive review of the literature with deliberately broad inclusion criteria, to get a wider view of this topic. Most studies in this review were small and there was marked heterogeneity between the study populations, notably in age, ethnicity and gestation. Levels of TnI were highly variable due to the different methods used, which were all lacking detail on analytical performance characteristics such as limits of quantitation, preventing direct comparison between studies. These substantial sources of heterogeneity prevent us from performing meaningful quantitative meta-analysis. We propose that this is representative of the diagnostic uncertainty around using cTn in pregnant women in clinical practice and justifies further investigation. We excluded women with measurements taken during or after labour or delivery, as this time is associated with complex, multifactorial physiological changes in addition to those described in the antenatal period [40]. While it would also be prudent to investigate cTn in the puerperium, it is beyond the scope of this study.

## Conclusion

TnI is elevated in women with pre-eclampsia but the mechanism and extent of this is unclear. For the first time, this review has evaluated the limited literature on using high-sensitivity cTn in pregnancy, which may be promising for future studies investigating cardiac disease in pregnant women. It may be appropriate to use the non-pregnant reference interval for TnI in women with an uncomplicated pregnancy, although the evidence from the existing literature is insufficient to robustly support this as the importance of gestational age is uncertain. TnT has rarely been investigated in pregnancy. There is an unmet clinical need for a suitably powered study to define the reference interval for cTn in uncomplicated pregnancy using highly sensitive methods. It would be prudent to extend this to include women with gestational hypertension and pre-eclampsia, ideally using placental biomarkers for case validation. We propose that this would constitute essential preliminary research, underpinning the safe assessment of pregnant women with suspected cardiovascular disease.

## Supporting information

**S1 Table. PRISMA checklist.**
(DOC)

**S2 Table. Search strategy.**
(DOCX)

**S3 Table. Quality assessment.**
(DOCX)

**S4 Table. Certainty of evidence.**
(DOCX)

## Author Contributions

**Conceptualization:** Samuel Dockree, Tim James, Manu Vatish.

**Data curation:** Samuel Dockree.

**Formal analysis:** Samuel Dockree, Jennifer Brook, Brian Shine, Tim James, Lauren Green, Manu Vatish.

**Investigation:** Samuel Dockree, Jennifer Brook, Brian Shine, Tim James, Lauren Green, Manu Vatish.

**Methodology:** Samuel Dockree, Tim James, Manu Vatish.

**Project administration:** Samuel Dockree.

**Software:** Samuel Dockree.

**Supervision:** Tim James, Manu Vatish.

**Writing – original draft:** Samuel Dockree, Tim James, Manu Vatish.

**Writing – review & editing:** Samuel Dockree, Jennifer Brook, Brian Shine, Tim James, Lauren Green, Manu Vatish.

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
