## [Decision Letter · Decision Letter 0]

21 Jan 2021

PONE-D-20-30756

Cardiac-specific troponins in uncomplicated pregnancy and pre-eclampsia: a systematic review

PLOS ONE

Dear Dr. Dockree,

Thank you for submitting your manuscript to PLOS ONE. After careful consideration, we feel that it has merit but does not fully meet PLOS ONE’s publication criteria as it currently stands. Therefore, we invite you to submit a revised version of the manuscript that addresses the points raised during the review process.

Please address the reviewer comments fully and paying particular attention to the discussion about appropriate control groups

I hope to see your manuscript in its revised form soon

We look forward to receiving your revised manuscript.

Kind regards,

Andrew Sharp, PhD

Academic Editor

PLOS ONE

Reviewers' comments:

Reviewer's Responses to Questions

**Comments to the Author**

1. Is the manuscript technically sound, and do the data support the conclusions?

Reviewer #1: Yes

2. Has the statistical analysis been performed appropriately and rigorously? 

Reviewer #1: Yes

3. Have the authors made all data underlying the findings in their manuscript fully available?

Reviewer #1: Yes

4. Is the manuscript presented in an intelligible fashion and written in standard English?

Reviewer #1: Yes

5. Review Comments to the Author

Reviewer #1: Abstract

10 studies only 8 summarised, what about others?

Is this sufficient evidence of cardiac stress or a leap?

Describe how to define 95% reference intervals specific to pregnancy, how do you know these are needed? Any studies examining difference between pregnant and non-pregnant controls?

Introduction

What is the biological basis and origin for circulating troponins? Please add this in brief.

Line 42. It would be better to state ‘used for’.

Line 46. Can you state any details for the reference population that may relate to pregnancy e.g. is there a female-specific reference range, what are the characteristics (age, pre-post menopausal) of included women. What is the magnitude of difference between when and women in the 99th centile cut-off?

Line 59 Add ‘further elevated in pre-eclampsia’

Line 62 onwards. I would advise stating or very briefly summarising the basis for the techniques, given the significance of the differences you describe.

Methods

Line 93 Would non-pregnant female controls not have been a second relevant group for comparison?

Line 97 Why exclude labour or immediate postpartum period as key risk times for cardiac events?

Were any limits based on study size applied and why not?

Results

The large volume of search results indicate the search could potentially have been refined, however studies selected for full review all appears relevant to aim.

Line 142 correct to quantification?

153 This statement needs revision. I think this is insufficient evidence for a dose response to blood pressure. BP is a manifestation of pre-eclampsia that does not always reflect the severity of the disease, other factors would need to be controlled to begin to comment on this appropriately.

Line 155 this is key information and should appear early in the results, please reposition. I think healthy non-pregnant women should have been included as an alternative comparator population. Can you confirm if they were included in the search (as you have identified this study).

Line 156 please clarify, results were all within the normal range (<99th centile) advised by the manufacturer. Also specify that this reference interval was constructed using data from healthy male (or male plus female) non-pregnant individuals. This is also key information as you are justifying need for pregnancy-specific reference intervals.

Line 160 onwards & Table 1. I would advise the statistical significance of these differences is specified in table 1.

Discussion (no line numbers to refer to in manuscript)

‘TnI in women with an uncomplicated pregnancy was similar to non-pregnant women but has rarely been compared directly’. This is important and needs elaboration. Please include information from other authors here.

I completely agree with your decision not to pool and meta-analyse in the context of high heterogeneity, this is good practice.

Conclusion

‘There is an unmet clinical need for a suitably powered study to define the reference interval for cTn in uncomplicated pregnancy using highly sensitive methods’ A short practical description of this would be beneficial.

6. PLOS authors have the option to publish the peer review history of their article (what does this mean?). If published, this will include your full peer review and any attached files.

Reviewer #1: No

---

## [Author Response · Author response to Decision Letter 0]

25 Jan 2021

Please see response to reviewers (numbered list of systematically addressed points)

---

## [Editor Report · Decision Letter 1]

17 Feb 2021

Cardiac-specific troponins in uncomplicated pregnancy and pre-eclampsia: a systematic review

PONE-D-20-30756R1

Dear Dr. Dockree,

We’re pleased to inform you that your manuscript has been judged scientifically suitable for publication and will be formally accepted for publication once it meets all outstanding technical requirements.

Kind regards,

Andrew Sharp, PhD

Academic Editor

PLOS ONE
---

## [Editor Report · Acceptance letter]

19 Feb 2021

PONE-D-20-30756R1 

Cardiac-specific troponins in uncomplicated pregnancy and pre-eclampsia: a systematic review 

Dear Dr. Dockree:

I'm pleased to inform you that your manuscript has been deemed suitable for publication in PLOS ONE. Congratulations! Your manuscript is now with our production department. 

Kind regards, 

on behalf of

Dr. Andrew Sharp 

Academic Editor

PLOS ONE